# Analysis of Transformers for Medical Image Retrieval

**Arvapalli Sai Susmitha**                                SUSMITHA@CSE.IITK.AC.IN
*IIT Kanpur, India*

**Vinay P. Namboodiri**                                VPN22@BATH.AC.UK
*University of Bath, UK*

**Editors:** Accepted for publication at MIDL 2024

## Abstract

This paper investigates the application of transformers to medical image retrieval. Although various methods have been attempted in this domain, transformers have not been extensively explored. Leveraging vision transformers, we consider co-attention between image tokens. Two main aspects are investigated: the analysis of various architectures and parameters for transformers and the evaluation of explanation techniques. Specifically, we employ contrastive learning to retrieve attention-based images that consider the relationships between query and database images. Our experiments on diverse medical datasets, such as ISIC 2017, COVID-19 chest X-ray, and Kvasir, using multiple transformer architectures, demonstrate superior performance compared to convolution-based methods and transformers using cross-entropy losses. Further, we conducted a quantitative evaluation of various state-of-the-art explanation techniques using insertion-deletion metrics, in addition to basic qualitative assessments. Among these methods, Transformer Input Sampling (TIS) stands out, showcasing superior performance and enhancing interpretability, thus distinguishing it from black-box models.

**Keywords:** Content-based medical Image Retrieval, Vision Transformers, Deep Learning, Contrastive Learning, Explainable AI.

## 1. Introduction

Over the past decade, the intersection of medical imaging and deep learning has witnessed significant advancements, addressing challenges in managing vast datasets, as highlighted by Hwang et al. (Hwang et al., 2012). Content-based medical image retrieval (MIR) has emerged as a crucial tool, aiding clinicians in recognizing related medical images and recalling prior cases during diagnosis (Agrawal et al., 2022). Traditionally, MIR has heavily relied on convolutional neural networks (CNNs), as evidenced by (Shetty et al., 2023) and (Qayyum et al., 2017). Despite their effectiveness in feature extraction and similarity identification, they pose challenges due to their inability to effectively capture long-range dependencies and the lack of interpretability, known as the "black-box" problem (Hu et al., 2022).

Our work addresses these challenges in medical image retrieval by adopting Vision Transformers (ViTs) (Dosovitskiy et al., 2021). ViTs, as explored in (El-Nouby et al., 2021), offer superior performance, excelling at capturing long-range dependencies and relationships between distant image regions through their multi-head attention mechanism (Zuo et al., 2022). Inspired by (El-Nouby et al., 2021) work we experiment with Vision Transformers employing contrastive learning and regularization, comparing their performance with convolutional baselines using both cross-entropy loss and contrastive loss under similar settings and also vision Transformers with cross-entropy loss. Our findings demonstrate the

superiority of Vision Transformers over these baselines for all datasets. Recognizing the significance of model explainability in clinical applications, We apply various state-of-the-art eXplainable AI (XAI) techniques tailored to Vision Transformers. Beyond simple qualitative comparisons, we conduct quantitative evaluations of saliency maps using insertion and deletion methods.

**The primary contributions of our work are as follows:**

- We provide a systematic analysis of vision transformer architectures (Section 4.2) and hyperparameters/variants (Section 4.3). Our findings consistently highlight the superiority of transformers with contrastive loss over convolutional baselines. These also outperform vision transformers and convolution baselines using cross-entropy loss across all datasets.

- Analysis of various state-of-the-art explanation techniques for Vision Transformers (Section 4.4). We quantitatively evaluate the resulting saliency maps using insertion and deletion methods.

## 2. Related Works

Content-based Medical Image Retrieval (MIR) plays a crucial role in enhancing diagnostic reliability for radiologists by retrieving pertinent medical cases that resemble a provided image. The early MIR methods relied on basic features and struggled to capture intricate relationships within medical images. The "semantic gap" between features and actual content led to inaccurate retrieval (Xuan et al., 1995) (Zhang et al., 2008). Convolutional neural networks (CNNs) revolutionized MIR and bridged the semantic gap by surpassing hand-crafted techniques (Sklan et al., 2015).

In the context of COVID, Kvasir, and ISIC datasets, studies (Tschandl et al., 2019), (Shetty et al., 2023), and (Agrawal et al., 2022) leveraged pre-trained CNN architectures like ResNet, VGG, Densenet for medical image retrieval. Recently,(Ahmed et al., 2023) proposed a novel relative difference-based similarity measure (RDBSM) for improved retrieval. Further, (Öztürk et al., 2023) have introduced an opponent class adaptive margin (OCAM) loss for S-bit hash code generation in image retrieval. CNN-based architectures have been extensively employed in the majority of studies published in the literature. However, the limitations of CNNs in capturing long-range dependencies prompted the exploration of vision transformers (ViTs) (El-Nouby et al., 2021). Inspired by the capabilities of ViT in general computer vision, we employ them with contrastive loss and differential entropy regularization for the task of medical image retrieval.

Few recent studies have explored various ViT-based approaches for Medical Image Retrieval. (Trinh and Nguyen, 2021) present a Mixer-MLP-based MIR for endoscopic images. (Thakrar et al., 2023) use a modified ViT for content-based image retrieval in chest X-rays, employing binary cross entropy and $L_1$ loss for training. (Gupta et al., 2023) propose dense-link-search for efficient nearest neighbours in medical image retrieval. (Manzari et al., 2023) have introduced MedViT, a hybrid model combining ViTs and CNNs for Medical Image Classification. However, these methods may lack interpretability, acting as black boxes in decision-making processes. (Hu et al., 2022) address interpretability concerns with their X-MIR method for CNNs, employing deep metric learning and similarity-based saliency maps for visual explanations of retrieved images. In our work, we undertake a benchmark-

ing study to analyse the various architectures and loss-functions and further analyse the interpretability of these methods.

Explaining Vision Transformers (ViTs) presents challenges, as traditional attention weight-based methods designed for CNNs are inadequate due to the distinctive nature of ViTs with multiple attention heads and encoder blocks (Serrano and Smith, 2019) (Stassin et al., 2023). To address this, various ViT-specific explainability methods are explored. Attention Rollout (Abnar and Zuidema, 2020), Chefer 2 (Chefer et al., 2021), Transition Attention Maps(TAMS) (Yuan et al., 2021), Bidirectional Transformers (BT) (Chen et al., 2022), ViT-CX (Xie et al., 2022), and Transformer Input Sampling (TiS) (Englebert et al., 2023) contribute uniquely to understanding ViT decision-making, providing insights into attention, gradients, and perturbation-based explanations.

## 3. Method

### 3.1. Transformers

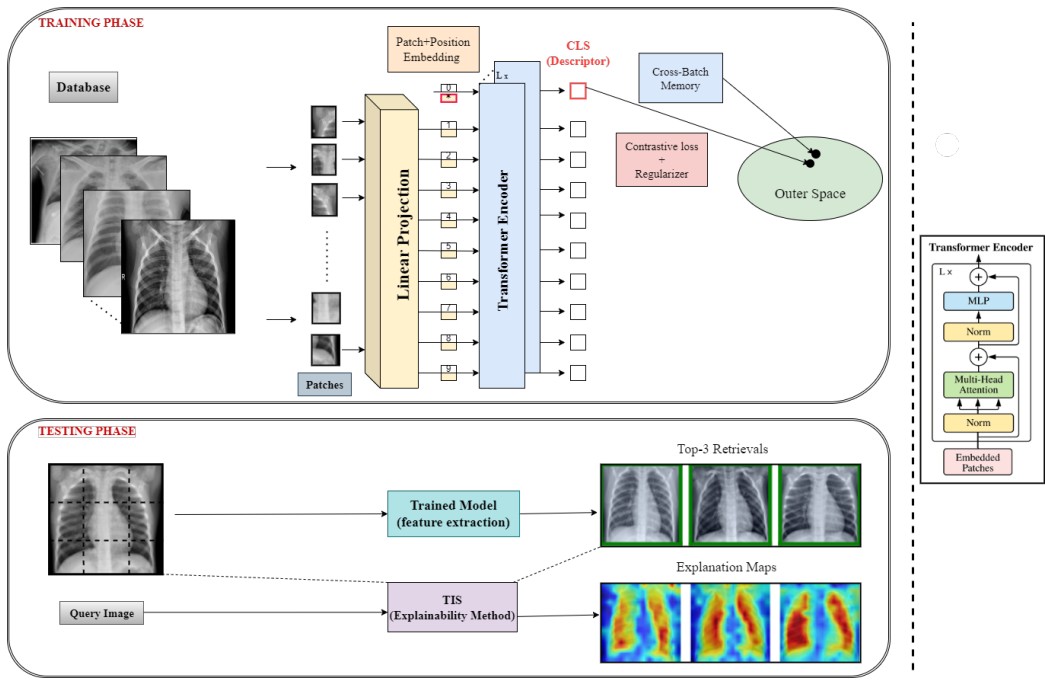

Figure 1: Transfomer model with Contrastive loss and Regularization for Medical Image Retrieval on COVID Dataset

The vision transformer (ViT), introduced by (Dosovitskiy et al., 2021), introduced tokenization of image patches for transformers, by transforming input images into a sequence of 2D patches (e.g., 16x16). These patches undergo a learnable linear projection, resulting in token embeddings. A special learnable [CLS] token is added at the sequence's beginning and serves as a global representation. The transformer encoder block consists of L layers, each of which is composed of two sub-layers: a multi-headed self attention (MSA) layer and

a multi-layer perceptron (MLP) layer. The resulting global representation, derived from the [CLS] token output, is used for subsequent processing.

As shown in Figure 1, we use transformers for content-based MIR.

In the training phase, we start with the pre-trained Vision Transformer (ViT) model and then fine-tuned for each dataset with metric learning, specifically employing a contrastive loss. Through this process, the model acquires feature embeddings i.e the cls token acts as the global image descriptor. The process maps similar images into a common feature space and cosine similarity is used to retrieve the closest images. A cross-batch memory(Wang and Isola, 2020) is used along with differential entropy regularization. The use of cross-batch memory enables to reduce the dependency on mini-batch for obtaining informative negative pairs. Using a cross-batch memory we are now able to increase the number of hard negatives without incurring significant computational overhead. In the testing phase, we deploy the trained ViT model to generate feature embeddings for each query image in a separate test dataset. These embeddings are then utilized to rank database images, enabling efficient medical image retrieval. Following this, performance metrics such as precision, recall, and F1 score are calculated to assess the efficacy of the retrieval process.

In our study, we explore three model variants: MIRDeiT_small, MIRViT_small, and MIRViT_base. MIRDeiT_small adopts the DeiT_small (Embedding length is 384)(Touvron et al., 2021) architecture with a 16x16 patch size and an image size of 224x224 pixels. Similarly, MIRViT_small follows the ViT_small architecture (Dosovitskiy et al., 2021) with matching patch and image dimensions (Embedding length is 384). In contrast, MIRViT_base is based on the larger ViT_base architecture (Embedding length is 768). All these models and the CNN models used for comparison are imported from the timm PyTorch library. The total loss $(L)$ used for training is a combination of the contrastive loss and the differential entropy regularization given as: $L = L_{\text{contr}} + \lambda L_{\text{KoLeo}}$

The contrastive loss $(L_{\text{contr}})$ encourages similarity among samples with the same label and dissimilarity among samples with different labels. Mathematically, it is expressed as:

$$L_{\text{contr}} = \frac{1}{N} \sum_i^N \left[ \sum_{j:y_i=y_j} \left[ 1 - z_i^T z_j \right] + \sum_{j:y_i \neq y_j} \left[ z_i^T z_j - \beta \right] \right]$$

Here, $z_i$ represents the $l_2$-normalized embedding vector of sample $i$, $N$ is the total number of samples, and $y_i$ is the label of sample $i$. The margin $\beta$ prevents the training signal from being dominated by easy negatives. Only negative pairs with a similarity higher than a constant margin $\beta$ contribute to the loss. The representations $z_i$ are assumed to be $l_2$-normalized, making the inner product equivalent to cosine similarity.

Simultaneously, $L_{\text{KoLeo}}$ is the differential entropy loss. It serves as a regularizer (Sablay-rolles et al., 2018) and is based on the (Kozachenko and Leonenko, 1987) differential entropy estimator. It prevents representations of different samples from being close by increasing their distance from positive examples and hard negatives. Mathematically, $L_{\text{KoLeo}}$ aims to maximize the distance between each point and its nearest neighbor:

$$L_{\text{KoLeo}} = -\frac{1}{N} \sum_{i=1}^N \log(\rho_i)$$

Here, $\rho_i$ represents the minimum distance between the embedding vector $z_i$ and any other embedding vector $z_j$ (where $j \neq i$). The regularization term is then used with a weighting coefficient $\lambda$.

During testing, each image in the test dataset, as depicted in Figure 1, acts as a query, with the top $k$ retrievals extracted for each query. The learned embedding network processes input images, denoted as $x$, producing embedding feature vectors $\mathbf{z}(x)$. This process applies to both query images ($q$) and retrieved images ($r$), resulting in feature vectors $\mathbf{z}_q$ and $\mathbf{z}_r$, respectively. To rank the retrieved images, a similarity score $s$ between $\mathbf{z}_q$ and $\mathbf{z}_r$ is computed using cosine similarity: $s(\mathbf{z_q}, \mathbf{z_r}) = \frac{\mathbf{z_q} \cdot \mathbf{z_r}}{\|\mathbf{z_q}\| \|\mathbf{z_r}\|}$. Standard image retrieval metrics, such as mean average precision (mAP), mean precision (mP@K), and Recall (R@K), are calculated for evaluation.

### 3.2. Explainability Methods

While vision transformers (ViTs) employ attention mechanisms, relying solely on raw attention is considered insufficient for comprehensive explanations. This raw attention overlooks the value component and emphasizes the query and key elements (Jain and Wallace, 2019; Serrano and Smith, 2019), This has led to the development of methods specifically made for ViTs. For instance, attention rollout (Abnar and Zuidema, 2020) combines attention heads and an identity matrix for residual connections. Chefer 2 (Chefer et al., 2021) offers a generic explanation method for transformers, using gradients and identity matrices for attention score computation. TAMs (Yuan et al., 2021) model representation evolution as a Markov chain, yielding class-specific explanations with integrated gradients. Bidirectional Transformer (BT) involves element-wise multiplication of Reasoning Feedback and Attention Perception (Chen et al., 2022), providing saliency maps for token (BT-T) and head (BT-H). ViT-CX (Xie et al., 2022) avoids direct dependence on attention weights, using perturbation masks derived from patch embeddings. Transformer Input Sampling (TiS) (Englebert et al., 2023) instead masks tokens before their introduction into a transformer, improving interpretability and reducing the number of tokens. We have compared a number of the above explainability methods to obtain explainable medical image retrieval and provide results for the same in section 4.4.

## 4. Results & Discussions

### 4.1. Datasets

In our comprehensive study, we utilize three datasets for medical image analysis. The curated COVID-19 Chest X-Ray Dataset (Sait et al., 2020) includes 1281 COVID-19 X-rays, 3270 Normal X-rays, and 4657 pneumonia X-rays (viral and bacterial). Our focus is on overall pneumonia classification. The ISIC Skin Lesion Dataset (Codella et al., 2018) comprises images of benign nevi, seborrheic keratosis, and melanoma (non-cancerous and malignant). This dataset has 2,750 images. The Kvasir-V2 dataset (Pogorelov et al., 2017) contains 8,000 annotated endoscopy images, categorized into eight classes by experienced endoscopists based on anatomical landmarks, pathological findings, or specific endoscopic procedures.

### 4.2. Evaluation of Architectures

We adopted the same training details as outlined in (El-Nouby et al., 2021), the optimization of these models employs the AdamW optimizer with a learning rate of $3 \times 10^{-5}$, weight

decay of $5 \times 10^{-4}$, and for 10k iterations. Contrastive loss margin ($\beta$) is set to 0.5. In the absence of regularization ($\lambda = 0$) and with differential entropy regularization, different variants of $\lambda$ ($\lambda = 0.3, 0.7$) are employed. Standard data augmentation techniques are applied, including resizing images to $256 \times 256$, random cropping to $224 \times 224$, and random horizontal flipping. The dynamic offline memory queue aligns with the dataset's size. In the case of cross-entropy, similar optimizer and iteration settings were used, applying basic cross-entropy loss for classification.

In our diverse evaluation across ISIC, COVID, and Kvasir datasets, as detailed in Table 1, we tested traditional CNNs (Densenet121, Resnet50), various vision transformers (DeiT_small, ViT_small, MedViT), and the MIRViT variants. MIRViT_small consistently outperforms CNNs, other vision transformers, and even MedViT(CNN-Transformer Hybrid) in medical image retrieval, demonstrating its efficacy.

In Table 1, to maintain conciseness, the results for ISIC are derived from $\lambda = 0.3$ for MIRViT_small and $\lambda = 0$ for MIRDeiT_small, while for Kvasir and COVID, MIRViT_small is based on $\lambda = 0.7$ and MIRDeiT_small is based on $\lambda = 0.3$ and $\lambda = 0.7$ respectively.The above results are based on three experimental runs. The complete set of results of the model combinations are provided in Figure 2.

Table 1: Medical Image retrieval results

| Dataset | Model | Loss | R@1 | R@5 | R@10 | mAP | mP@1 | mP@5 | mP@10 |
|---|---|---|---|---|---|---|---|---|---|
| ISIC | Densenet121 | Contrastive | 64.50 | 92.50 | 95.67 | 58.38±0.01 | 64.50 | 62.37 | 62.57 |
| | Resnet50 | | 65.33 | 92.33 | 97.00 | 57.72±0.02 | 65.33 | 65.13 | 63.90 |
| | MIRViT_small | | 74.17 | 88.17 | 91.33 | **70.96±0.01** | 74.17 | 73.80 | 73.72 |
| | MIRDeiT_small | | 71.83 | 89.5 | 94.67 | 68.44±0.02 | 71.83 | 71.07 | 71.55 |
| | MedViT_S | | 59.00 | 90.00 | 97.17 | 51.10±0.01 | 59.00 | 55.93 | 54.65 |
| | DeiT_small | Cross Entropy | 71.33 | 90.50 | 95.17 | 63.32±0.02 | 71.33 | 70.87 | 70.10 |
| | ViT_small | | 69.00 | 90.50 | 95.83 | 60.11±0.0 | 69.00 | 65.67 | 64.57 |
| | Densenet121 | | 60.50 | 91.00 | 96.17 | 58.80±0.01 | 60.50 | 60.87 | 60.18 |
| | Resnet50 | | 67.33 | 90.50 | 97.17 | 53.99±0.01 | 67.33 | 64.73 | 63.15 |
| COVID | Densenet121 | Contrastive | 96.20 | 98.8 | 99.19 | 94.62±0.01 | 96.20 | 95.87 | 95.73 |
| | Resnet50 | | 94.24 | 98.53 | 99.02 | 91.33±0.01 | 94.24 | 93.86 | 93.91 |
| | MIRViT_small | | 97.72 | 98.26 | 98.53 | **96.96±0.01** | 97.72 | 97.52 | 97.51 |
| | MIRDeiT_small | | 96.80 | 98.37 | 98.80 | 96.48±0.02 | 96.80 | 96.74 | 96.65 |
| | MedViT_L | | 89.95 | 98.04 | 98.75 | 83.29±0.01 | 89.95 | 89.57 | 89.24 |
| | DeiT_small | Cross Entropy | 95.11 | 98.53 | 98.86 | 92.93±0.01 | 95.11 | 94.89 | 94.65 |
| | ViT_small | | 93.05 | 97.94 | 98.59 | 93.24±0.01 | 93.05 | 93.44 | 93.30 |
| | Densenet121 | | 87.72 | 97.07 | 98.59 | 80.38±0.01 | 87.72 | 87.19 | 86.69 |
| | Resnet50 | | 92.07 | 97.66 | 98.59 | 82.35±0.02 | 92.07 | 90.87 | 90.20 |
| Kvasir | Densenet121 | Contrastive | 88.83 | 97.58 | 98.25 | 83.89±0.01 | 88.83 | 88.98 | 88.69 |
| | Resnet50 | | 90.42 | 97.46 | 98.67 | 84.85±0.02 | 90.42 | 89.75 | 89.63 |
| | MIRViT_small | | 93.33 | 96.92 | 97.54 | **90.16±0.01** | 93.33 | 92.87 | 92.84 |
| | MIRDeiT_small | | 92.21 | 96.92 | 97.96 | 90.11±0.01 | 92.21 | 92.50 | 92.53 |
| | MedViT_T | | 68.33 | 95.29 | 98.50 | 51.41±0.01 | 68.33 | 64.65 | 62.82 |
| | DeiT_small | Cross Entropy | 91.46 | 97.38 | 98.25 | 88.15±0.02 | 91.46 | 91.62 | 91.39 |
| | ViT_small | | 86.50 | 96.71 | 98.00 | 79.33±0.01 | 86.50 | 86.59 | 86.24 |
| | Densenet121 | | 56.71 | 89.08 | 95.21 | 26.23±0.01 | 56.71 | 52.62 | 50.28 |
| | Resnet50 | | 66.96 | 92.17 | 95.88 | 40.59±0.01 | 66.96 | 63.74 | 61.42 |

MIRViT_small's superiority is evident, for example for the ISIC dataset, achieving higher recall and a significant 13.24% increase in mAP compared to Resnet50. It also outshines vision transformers trained with cross-entropy loss, emphasizing its precision in top-k scenarios, with a notable 7.64% surge in mAP compared to DeiT_small. Similarly, across the COVID and Kvasir datasets, MIRViT_small outperforms convolution-based methods and

transformers using cross-entropy losses, showcasing versatility in diverse medical imaging contexts. MedViT, while excelling in classification, lags in retrieval. In summary, vision transformers, especially MIRViT_small, exhibit good potential in medical image retrieval. Their consistent superiority in recall, precision, and mAP underscores their effectiveness. Use of contrastive loss, with or without differential entropy regularization is beneficial.

### 4.3. Evaluation of Hyperparameters and Transformer Variants

The assessment of image retrieval results in Figure 2 consistently shows the dominance of MIRViT_small over MIRDeiT_small and MIRViT_base across diverse datasets, including ISIC, COVID, and Kvasir. In the ISIC dataset, MIRViT_small consistently outperforms other configurations, delivering optimal performance at $\lambda = 0.3$. For the COVID and Kvasir datasets, MIRViT_small at $\lambda = 0.7$ achieves the highest mAP, closely followed by $\lambda = 0.3$ with a small difference. Notably, MIRViT_base does not emerge as the top performer for all datasets, indicating that a 384 embedding length performs well for retrieval on these medical datasets.In conclusion, MIRViT_small with $\lambda = 0.3$ emerges as an optimal choice for medical image retrieval tasks.

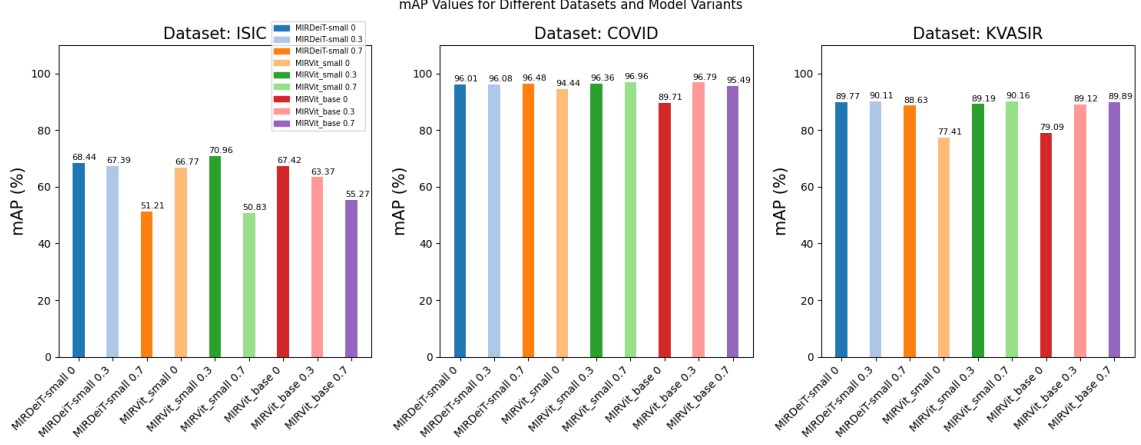

Figure 2: mAP Values for Different Datasets and Model Variants

### 4.4. Evaluation of Image Retrieval Explanations:

To assess visual explanations, we employ insertion and deletion casual metrics in image retrieval, gauging how well the generated explanations capture the causes behind predictions. We measure changes in image similarity as a result of changes to the retrieved image. The insertion metric measures increased image similarity by starting with a blurred version of the original retrieved image and gradually revealing pixels from highest to lowest relevance. Conversely, the deletion metric assesses the decline in image similarity we gradually mask out pixels on the retrieved image with a constant gray value from highest relevance to lowest based on the computed saliency map. We then compute the similarity score s between the query image q and perturbed versions of the retrieved image $\hat{r}$ (either in the form of insertion onto a blurred image or deletion using a constant gray value) $s(\mathbf{z_q}, \mathbf{z_{\hat{r}}}) = (\max(0, \frac{\mathbf{z_q} \cdot \mathbf{z_{\hat{r}}}}{\|\mathbf{z_q}\| \|\mathbf{z_{\hat{r}}}\|}))$.

To rectify non-negative outputs, all similarity values are adjusted to a minimum of zero. The area under the curve (AUC) is used to measure the effectiveness of saliency maps. Higher AUC values are preferred for insertion, and lower AUC values are desirable for deletion.

We implemented all explainability methods using the same hyperparameters as in (Englebert et al., 2023). In the saliency maps, vibrant (red) regions signify the primary focus, while cooler (blue) areas have less impact. The maps are rescaled using bilinear interpolation to align with the input image resolution.

Chefer2 and TIS consistently emerge as strong performers Table 2, shows top-tier scores across datasets and metrics. However, the qualitative evaluation through visual explanation maps shown in Figure 3 suggests that TIS is qualitatively better. On the other hand,

Table 2: AUC values for insertion and deletion metrics on MIRViT_small with various explanation techniques

| Dataset | Metric | bth | btt | chefer2 | rollout | tam | tis | vitcx |
|---------|--------|-----|-----|---------|---------|-----|-----|-------|
| ISIC | Insertion | 0.79 | 0.79 | 0.79 | 0.78 | 0.79 | 0.78 | 0.72 |
| | Deletion | 0.46 | 0.43 | 0.41 | 0.45 | 0.44 | 0.41 | 0.53 |
| COVID | Insertion | 0.67 | 0.66 | 0.70 | 0.69 | 0.66 | 0.67 | 0.62 |
| | Deletion | 0.45 | 0.47 | 0.42 | 0.44 | 0.47 | 0.46 | 0.51 |
| Kvasir | Insertion | 0.72 | 0.72 | 0.74 | 0.74 | 0.71 | 0.72 | 0.68 |
| | Deletion | 0.46 | 0.46 | 0.40 | 0.42 | 0.48 | 0.42 | 0.49 |

Rollout, while maintaining competitive AUC values, displays a tendency towards higher deletion scores and falls short in delivering detailed visual explanations. Other methods in the comparison, such as BTT, BTH, VitCX, and TAM, exhibit inconsistency, excelling in one metric while lagging in another. For detailed results please refer to Appendix A.

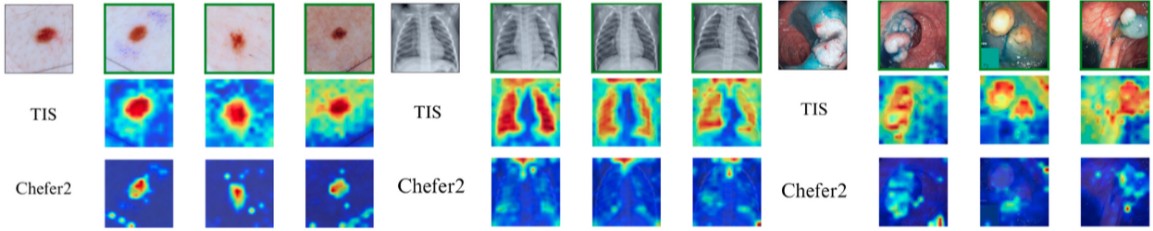

Figure 3: MIRViT_small top-3 Retreivals and Explanation maps on all Datasets

## 5. Conclusion

In conclusion, our study presents an analysis of various architectures and parameters for transformers, along with the evaluation of explanation techniques. MIRViT small emerges as the top-performing model across varied datasets ISIC, COVID, and Kvasir. Our exploration of loss functions underscores the limitations of simple cross-entropy and the effectiveness of the contrastive approach. Further, our analysis of state-of-the-art eXplainable AI methods suggests Transformer Input Sampling (TIS) as being better. In future, we intend to further explore various advances for these algorithms for the medical image retrieval task.

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

**Appendix A.**

The Figure 4 visually demonstrates the intermediate steps of the Insertion-Deletion metrics using the TIS explainability method on the MIRViT_small model. The deletion process is performed on a Kvasir image, while the insertion process is applied to a COVID chest X-ray image. Both processes effectively highlight the evolving significance of pixels in the explanation sequences. The value **p** represents the distance between the query image and

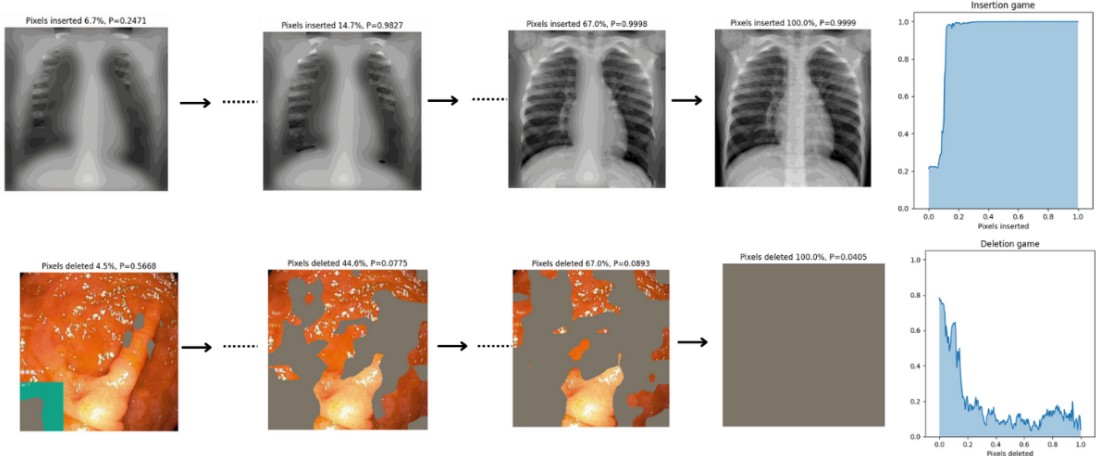

Figure 4: TIS Insertion & Deletion Steps

the intermediate perturbed versions of the retrieved image. As anticipated, the **p** value decreases during deletion as pixels are gradually removed and increases during insertion with the progressive addition of pixels. The area under the curve for these graphs (AUC) serves as the quantitative measure of the effectiveness of the saliency maps.

In Figure 5, we present top-3 retrievals using various explainability techniques for the ISIC dataset. Subfigure a) illustrates a query image with accurate retrievals (indicated by a green border). Saliency maps distinctly concentrate on lesion regions, providing meaningful insights. On the other hand, Subfigure b) showcases two incorrect retrievals (marked with a red border), where the saliency maps tend to focus on areas around the lesion rather than precisely on the lesion itself. In the first incorrect retrieval, TIS exhibits a focus on some scale at the bottom of the image, while in the subsequent incorrect retrieval, it predominantly concentrates around the lesion, focusing on non-lesion regions.

Additionally, Figure 6 displays an example query image alongside its top-3 retrievals for the COVID and Kvasir datasets, where TIS explanations are visually better than other methods. Figure 7 further illustrates TIS explanation maps for the top-3 retrievals, each representing one example image per class, demonstrating the effectiveness of saliency maps in highlighting relevant regions.

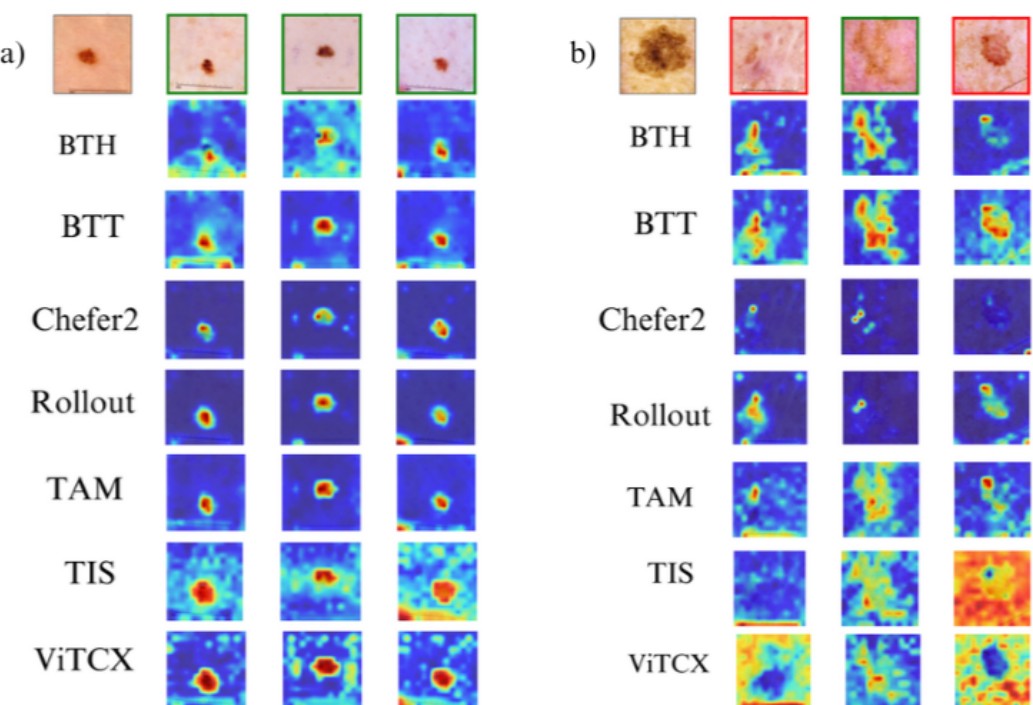

Figure 5: Top-3 Retrieval Explanations for Two Images from ISIC Dataset Using Different Explanation Methods.

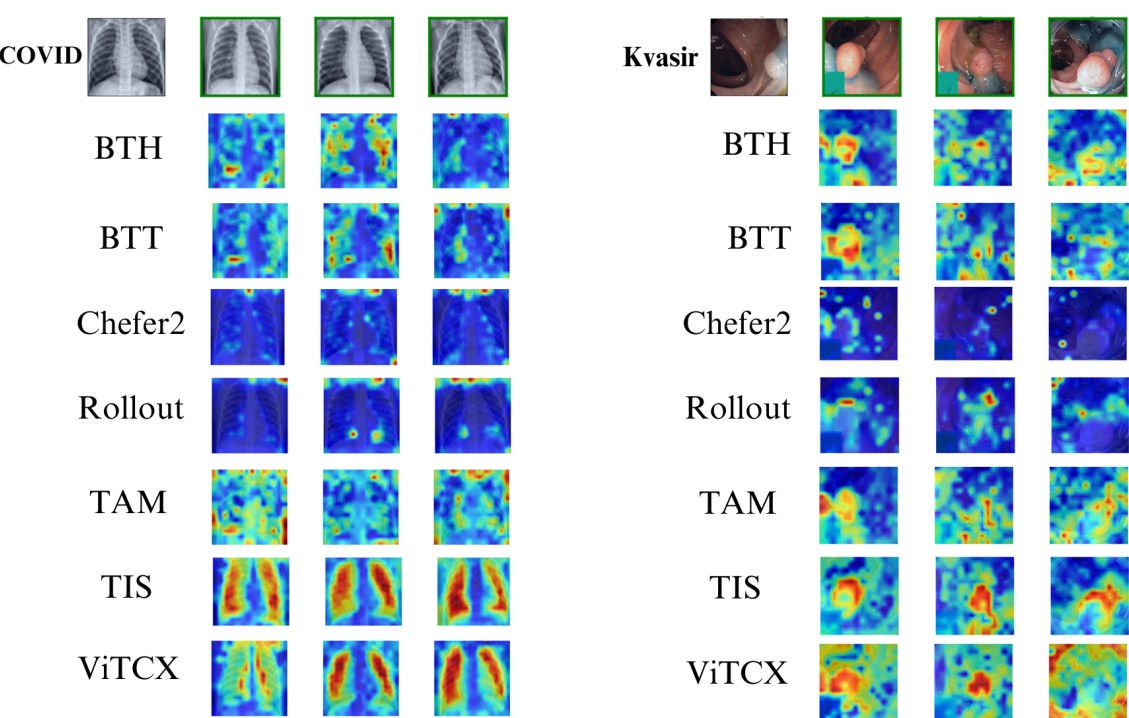

Figure 6: Top-3 Retrieval Explanations for One Example Image from COVID and Kvasir Using Different Explanation Methods.

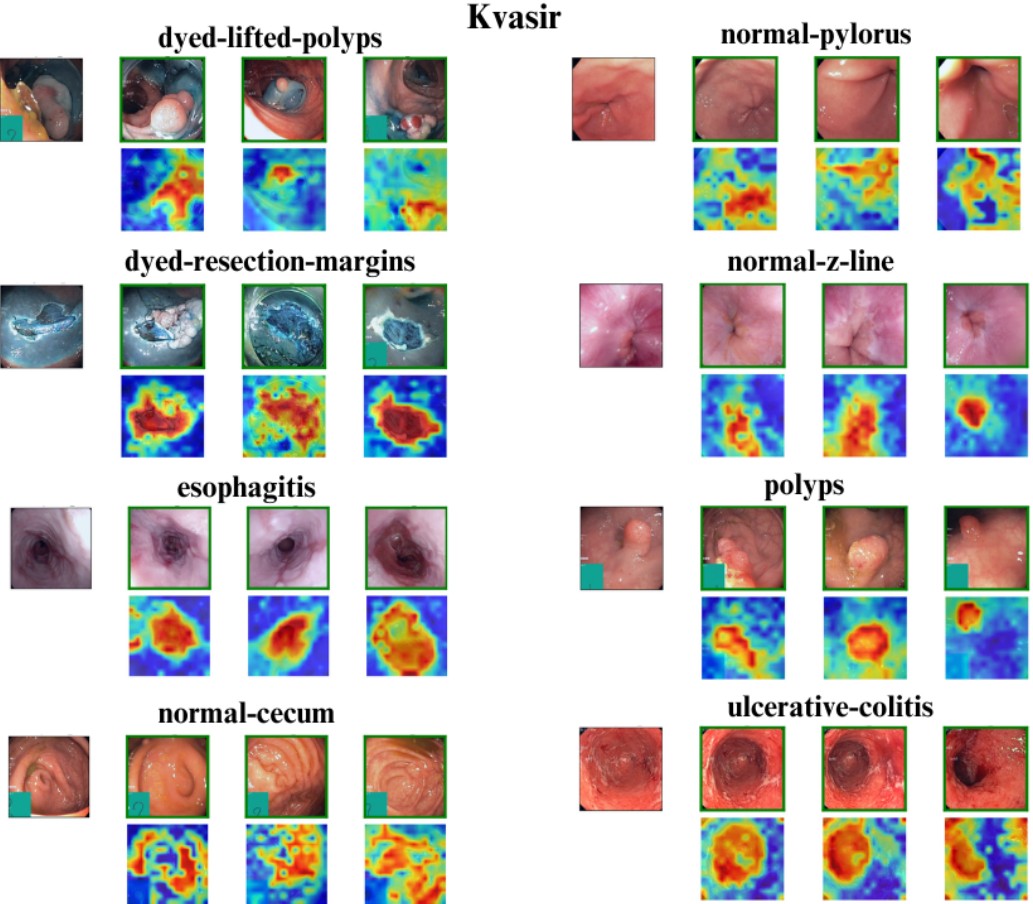

Figure 7: Top-3 Retrieval Explanations for an Example Image from Each of the 8 Classes in Kvasir using TIS.

Table 3: Model and the Number of Learnable Parameters

| Model | # Parameters |
|---|---|
| MedViT_small (CNN-Transformer Hybrid) | 31.14M |
| MedViT_base (CNN-Transformer Hybrid) | 44.41M |
| MedViT_large (CNN-Transformer Hybrid) | 57.68M |
| ResNet50 | 23.51M |
| DenseNet121 | 6.95M |
| MIRViT_small | 21.67M |
| MIRViT_base | 85.80M |
| MIRDeiT_small | 21.67M |

As can be observed from the table 3, the MIRViT_small and MIRDeiT_small have fewer parameters than ResNet50 and have improved performance. DenseNet model is more compact, but in general, the parameter settings for other small and base models are comparable.

