# OpenReview forum: "Analysis of Transformers for Medical Image Retrieval"
_MIDL.io/2024/Conference — MIDL 2024 Poster_

### Official Review · Reviewer_eaBC · 2024-02-28

**Confidence:** 4
**Preliminary Rating:** 2
**Final Rating:** 3.5

**Summary:**

The study explores the use of transformers in medical image retrieval. Results from experiments conducted on three datasets indicate that a transformer trained with contrastive loss surpasses convolution-based methods and transformers trained with cross-entropy loss. Additionally, a quantitative assessment of several state-of-the-art explanation techniques is performed, revealing superior performance of TIS.

**Strengths:**

The motivation is clear and easy to understand.

The paper evaluates major training methods.

The experiment dataset is comprehensive, covering 3 modalities.

The model interpretation study is attractive and useful.

**Weaknesses:**

• Despite being an analysis paper focused on the backbone, the reviewer remains apprehensive about its novelty.

• The pretraining and finetuning strategy lacks clarity.

• Figure 1 lacks clarity in its presentation. The inclusion of both training and testing phases is confusing.

• The presentation of results in Table 1 is unclear, and the comparison between different backbones may not be equitable.

• More backbones such as Swin Transformer and CNN-Transformer hybrid backbones shall be included and evaluated.

**Detailed Comments:**

•	The paper should include a more extensive discussion on novelty. While it positions itself as an analysis paper, it would enhance its content to explore novelty regarding pretraining/finetuning strategies, evaluation methods, and any significant discoveries.

•	In the method section, the paper states, "We then fine-tune the transformer." Reviewers have noted that the pre-pretraining step, which involves pretraining on ImageNet 1k, is mentioned in the following paragraph. It would be preferable to adjust the order of this presentation for clarity.

•	Based on Figure 1, it appears that the CLS token embedding is utilized for learning tasks. However, it remains unclear which image token embedding is employed during testing for the MIR task.

•	As described in the method section, the combination of contrastive loss and differential entropy regularization loss is employed for fine-tuning. Which specific loss corresponds to this combination in Table 1?

•	The comparison presented in Table 1 may lack fairness due to the discrepancy in the total number of learnable parameters.

**Justification Of Final Rating:**

I would be grateful for the authors' feedback. most of my concerns have been addressed, with the exception of the issue regarding novelty. I am confident that the readability of the revised version has been improved, and I believe that the community stands to gain valuable insights from this work. Therefore, I have decided to adjust my rating accordingly.

**Justification Of The Preliminary Rating:**

The paper demonstrates clear motivation and maintains an easily understandable presentation. It offers a comprehensive evaluation across multiple datasets. However, a significant flaw exists within the paper, leading to a preliminary rating based on its limited novelty and unclear presentation of the method and evaluation results.

**Questions To Address In The Rebuttal:**

Please provide a detailed discussion of this paper’s novelty, and please respond to the questions in the Comments section.

---

> ### Author Response · Authors · 2024-03-17
>
> We thank the reviewer for their comments and suggestions. Below are the clarifications for their comments:
>
> 1) The primary contributions of the paper include the analysis of transformers with contrastive loss for medical image retrieval and the evaluation of explanation methods. While these methods are well-established for classification on datasets like ImageNet, their application in medical image retrieval settings is not so well analyzed. With this work, we seek to provide strong comparable baselines that can be further used by the community. As is evident, we are not advocating a new loss function or architecture in this paper. However, we believe the community does benefit from the benchmarks and analysis of techniques that we provide. We intend to provide these models, code, and analysis techniques as open source to enable the community to benefit further from this work on acceptance.
>
> 2) The suggestions by the reviewer have been incorporated into the paper (page 4 highlighted in red). Thanks for the suggestions.
>
> 3) We adopt the CLS token embedding for both training and testing. Specifically, the CLS token for each query image in a separate test dataset is computed and used to compute the cosine distance to the retrieval set of images.
>
> 4) The complete set of results for the combinations is provided in Figure 2. In Table 1, to maintain conciseness, the results for ISIC are derived from λ=0.3 for MIRViT_small and λ=0 for MIRdeiT_small, while for KVASIR and COVID, MIRViT_small is based on λ=0.7 and MIRdeiT_small is based on λ=0.3 and λ=0.7 respectively. It is worth noting that, as depicted in Figure 2, adopting λ=0.3 across all settings would not be detrimental.  In general, we would recommend using cross-validation for practical use in clinical settings. We have expanded our explanation on page 6 (highlighted in red) for better understanding.
>
> 5) As recommended by the reviewer, we have now included the number of parameters for each architecture in the appendix, page 13 as Table 3. As can be observed from the table, the MIRViT_small and MIRdeiT_small have fewer parameters than ResNet50 and have improved performance. The DenseNet model is more compact, but in general, the parameter settings for other small and base models are comparable. These results are consistent with classification performance results.
>
> Also as recommended by the reviewer, it would indeed be beneficial to incorporate comparisons with several additional transformer baselines, including the Swin Transformer. However, we have access to limited computational resources and therefore, if we obtain access to sufficient computational resources, we will definitely include Swin transformer and additional transformer baselines. We believe that the present usage of ViT and deiT does provide an indication of the performance capabilities of transformers for the task of medical image retrieval.

---

### Official Review · Reviewer_Md37 · 2024-02-28

**Confidence:** 5
**Preliminary Rating:** 4
**Final Rating:** 4

**Summary:**

This paper investigates various vision transformers that incorporate contrastive learning for medical image retrieval. The authors conducted a systematic analysis of vision transformers and their hyperparameters, showcasing the superiority of transformers with contrastive loss over convolutional baselines. They demonstrated that the contrastive loss aids in outperforming vision transformers and convolutional baselines that rely solely on cross-entropy loss. The authors evaluated their methods across three distinct and diverse datasets: ISIC 2017, COVID-19 chest X-ray, and Kvasir.

**Strengths:**

1. The paper is well-written, well-organized, and easy to follow.
2. The paper offers an in-depth systematic analysis of various vision transformer architectures, comparing contrastive loss and cross-entropy loss for the medical image retrieval task.
3. The experiments demonstrate the superiority of contrastive loss over cross-entropy loss.
4. The authors clearly articulate the distinction between cross-entropy loss and their novel contrastive loss, which includes differential entropy regularization.
5. The authors provide a detailed appendix with additional information on data preprocessing and methods.

**Weaknesses:**

1. The authors could extend their investigation to include different variations of Swin Transformers for the Medical Image Retrieval task.
2. It would be intriguing to observe the impact of the formulated contrastive loss on Swin Transformers
3. It is not evident from the manuscript how many times each experiment was repeated. Demonstrating statistical significance across multiple runs would enhance the paper.
4. The authors could illustrate the convergence speed of the model using contrastive loss versus cross-entropy loss;

**Detailed Comments:**

Please see the points included in the section "Weaknesses".

**Justification Of Final Rating:**

The authors have satisfactorily addressed all of my questions and have made necessary adjustments in their experiments. However, I believe that exploring other variants of Swin Transformers would significantly enhance the strength of their paper.

**Justification Of The Preliminary Rating:**

The work presented in the paper is intriguing. Overall, the paper is well written and organized. The authors have provided a clear explanation of their experimental setup and have shown promising results by employing the contrastive loss in various Vision Transformer architectures. They evaluated their method across three different and diverse datasets and successfully demonstrated performance improvements using their approach.

**Questions To Address In The Rebuttal:**

Please see the points included in the section "Weaknesses".

---

> ### Author Response · Authors · 2024-03-17
>
> We thank the reviewer for their comments and suggestions and their appreciation of the work. We provide clarifications below for the points raised and we have also amended the paper based on these points.
>
> 1) Thanks to the reviewer for appreciating the work and recommending the analysis using Swin transformers. We intend to incorporate this analysis in the future. At present, due to computational bottlenecks, we have not been able to do the same.
>
> 2) Yes, indeed, we find the idea interesting. As mentioned, we plan to explore the analysis using Swin transformers in the future. We appreciate the suggestion to compare the contrastive loss using Swin transformers.
>
> 3) Thank you for raising this point. We have run each experiment three times. In the revised version we provide the mean and standard deviation for each experiment in Table 1. As can be observed there is less standard deviation.
>
> 4) The models using cross-entropy loss have converged around 2000 epochs and contrastive loss have converged around 6000 epochs.

---

### Official Review · Reviewer_xWw6 · 2024-02-29

**Confidence:** 4
**Preliminary Rating:** 3
**Final Rating:** 3.5

**Summary:**

The paper performs a benchmarking analysis of three Vision Transformer (ViT) models for medical image retrieval across three datasets encompassing chest x-ray, dermoscopy, and endoscopy images. It compares contrastive loss against cross-entropy loss and highlights the superiority of transformers trained with contrastive loss. Furthermore, the study delves into seven model interpretability techniques, providing both quantitative and qualitative evaluations, ultimately identifying Transformer Input Sampling as the most effective approach.

**Strengths:**

1. The paper evaluates three ViT models for medical image retrieval across three datasets encompassing different modalities.

2. The paper investigates seven model interpretability techniques and identifies the most effective approach.

**Weaknesses:**

1. Figure 1 illustrates the input of both the query image and the database image to ViT simultaneously, yet lacks explicit explanation, leading to confusion about the workflow.

2. The method description should provide further clarification on the cross-batch memory. Specifically, how does this mechanism facilitate the mapping of similar images into a common feature space?

3. Section 4.3 concludes that MIRViT_small with λ = 0.3 emerges as an optimal choice for medical image retrieval tasks. However, this setup does not consistently yield the best performance across all three tasks, rendering the argument less tenable.

4. The introduction to the insertion and deletion causal metrics lacks clarity regarding why and how these metrics quantify the explainability of the techniques.

**Detailed Comments:**

N/A

**Justification Of Final Rating:**

I am very grateful to the authors for responding positively to my review comments and making the appropriate changes in the original manuscript. I believe the revised version has been improved, and I updated my rating accordingly.

**Justification Of The Preliminary Rating:**

This study serves as a benchmark, evaluating three ViT models for medical image retrieval across three datasets. While the experimental results are relatively comprehensive, the clarity of the methodology and the depth of discussion derived from the results are lacking.

**Questions To Address In The Rebuttal:**

The question listed in the Weaknesses needs to be clarified.

---

> ### Author Response · Authors · 2024-03-17
>
> Thanks to the reviewer for their comments and suggestions. We have revised the paper based on the same and we provide the response below for the comments:
>
> 1) As recommended by the reviewer, we have now revised Figure 1 to clearly depict both the training and testing phases separately. The updated figure is now presented on page 3. Moreover, we've added further explanations about the training and testing processes, highlighted in red on page 4.
>
> 2) As requested by the reviewer, we have expanded the explanation of cross-batch memory usage on page 4. Essentially, incorporating cross-batch memory enables us to reduce the dependency on mini-batch for obtaining informative negative pairs. With the implementation of cross-batch memory, we can now increase the number of hard negatives without incurring significant computational overhead.
>
> 3) As the reviewer has correctly identified, λ = 0.3 is not always the optimal choice. The other parameter that did do well was λ =0.7 which however did not do well on the ISIC dataset. As a recommendation for practical usage in a clinical setting, we would always advocate a cross-validation of the parameter setting on the desired actual dataset and the analysis provided is mainly an indicative recommendation for initial consideration.
>
> 4) As recommended by the reviewer, we have now expanded the explanation of these metrics on page 7 highlighted in red. The insertion and deletion metrics involve intervention-based analysis. We modify the samples and observe the difference in performances. For instance,  when we blur part of an image, we can gauge the significance of the unblurred regions for retrieval. Qualitative examples of the resulting maps are presented in Figure 4 on page 9. While the analysis methods  were originally introduced for classification in Petsiuk, Vitali et al. “RISE: Randomized Input  Sampling for  Explanation of Black-box Models.” ArXiv abs/1806.07421 (2018)., we illustrate its utility for retrieval analysis.

---

### Decision · Program_Chairs · 2024-04-05

Accept (Poster)